# 'Don't Get Too Technical with Me':
# A Discourse Structure-Based Framework for Science Journalism

**Ronald Cardenas[1], Bingsheng Yao[2], Dakuo Wang[3], and Yufang Hou[4]**

[1]University of Edinburgh, [2]Rensselaer Polytechnic Institute
[3]Northeastern University, [4]IBM Research Europe, Ireland
ronald.cardenas@ed.ac.uk, yaob@rpi.edu
d.wang@northeastern.edu, yhou@ie.ibm.com

## Abstract

*Science journalism* refers to the task of reporting technical findings of a scientific paper as a less technical news article to the general public audience. We aim to design an automated system to support this real-world task (i.e., **automatic science journalism**) by 1) introducing a newly-constructed and real-world dataset (SCITECHNEWS), with tuples of a publicly-available scientific paper, its corresponding news article, and an expert-written short summary snippet; 2) proposing a novel technical framework that integrates a paper's discourse structure with its metadata to guide generation; and, 3) demonstrating with extensive automatic and human experiments that our framework outperforms other baseline methods (e.g. Alpaca and ChatGPT) in elaborating a content plan meaningful for the target audience, simplifying the information selected, and producing a coherent final report in a layman's style.

## 1 Introduction

*Science journalism* refers to producing journalistic content that covers topics related to different areas of scientific research (Angler, 2017). It plays an important role in fostering public understanding of science and its impact. However, the sheer volume of scientific literature makes it challenging for journalists to report on every significant discovery, potentially leaving many overlooked. For instance, in the year 2022 alone, $185,692$ papers were submitted to the preprint repository *arXiv.org* spanning highly diverse scientific domains such as biomedical research, social and political sciences, engineering research and a multitude of others[1]. To this date, PubMed contains around $345,332$ scientific publications about the novel coronavirus Covid-19[2], nearly 1.6 times as many as those produced in 200 years of work on influenza.

---

[1]https://info.arxiv.org/about/reports/2022_arXiv_annual_report.pdf
[2]https://www.ncbi.nlm.nih.gov/research/coronavirus/

The enormous quantity of scientific literature and the huge amount of manual effort required to produce high-quality science journalistic content inspired recent interest in tasks such as generating blog titles or slides for scientific papers (Vadapalli et al., 2018; Sun et al., 2021), extracting structured knowledge from scientific literature (Hou et al., 2019; Mondal et al., 2021; Zhang et al., 2022), simplifying technical health manuals for the general public (Cao et al., 2020), and creating plain language summaries for scientific literature (Dangovski et al., 2021; Goldsack et al., 2022).

Our work focuses on generating simplified journalistic summaries of scientific papers for the non-technical general audience. To achieve this goal, we introduce a new dataset, SCITECHNEWS, which pairs full scientific papers with their corresponding press release articles and newswire snippets as published in *ACM TechNews*. We further carry out in-depth analysis to understand the journalists' summarization strategies from different dimensions (Section 3.2). Then, we explore novel model designs to generate short journalistic summaries for scientific papers. Unlike previous studies that model this problem as a "flat" sequence-to-sequence task and ignore crucial metadata information of scientific papers (Dangovski et al., 2021; Goldsack et al., 2022), we propose a **technical framework** that integrates author and affiliation data as they are important information in scientific news articles. Furthermore, we encode each sentence with its corresponding **discourse rhetoric role** (e.g., *background* or *methods*) and apply a hierarchical decoding strategy to generate summaries. As illustrated in Figure 1, our trained decoding model first generates a content plan, which is then employed to guide the model in producing summaries that adhere to the plan's structure.

In summary, our main contributions are twofold. First, we construct a new open-access high-quality dataset for *automatic science journalism*

| Input article and Metadata |
| --- |
| [AUTHOR] ron shmelkin \| tel aviv university [AUTHOR] ... [BACKGROUND] a master face is a face image that passes facebased identity - authentication for a large portion of the population . ... [CONCLUSIONS] this is demonstrated for multiple face representations and explored with multiple , state - of - the - art optimization methods . |

| Content Plan and Target Summary |
| --- |
| **[PLAN]** [AUTHOR] [BACKGROUND] \| [BACKGROUND] [RESULTS] \| [BACKGROUND] [METHODS] [RESULTS] \| [AUTHOR] [METHODS] [RESULTS] **[SUMMARY]** computer scientists at israel's tel aviv university ( tau ) say they have developed a "master face" method for circumventing a large number of facial recognition systems , by applying artificial intelligence to generate a facial template . the researchers say the technique exploits such systems ' usage of broad sets of markers to identify specific people ; producing facial templates that match many such markers essentially creates an omni - face that can bypass numerous safeguards . the researchers created the master face by plugging an algorithm into a generative adversarial network that builds digital images of artificial human faces . the tau team said testing showed the template was able unlock over 20 % of the identities in an open source database of 13,000 facial images operated by the university of massachusetts . |

Figure 1: An example of a scientific article enriched with metadata and scientific rhetoric roles, along with its content plan and target press summary. Colors relate to the plan at the sentence level.

that covers a wide range of scientific disciplines. Second, we propose a novel approach that learns the discourse planning and the writing style of journalists, which provides users with fine-grained control over the generated summaries. Through extensive automatic and human evaluations (Section 6), we demonstrate that our proposed approach can generate more coherent and informative summaries in comparison to baseline methods, including zero-shot LLMs (e.g., ChatGPT and Alpaca). In principle, our framework can assist journalists to control the narrative plans and craft various forms of scientific news summaries efficiently. We make the code and datasets publicly available at `https://github.com/ronaldahmed/scitechnews`.

## 2 Related work

**Existing Datasets.** There are a few datasets for generating journalistic summaries for scientific papers. Dangovski et al. (2021) created the Science Daily dataset, which contains around 100K pairs of full-text scientific papers and their corresponding Science Daily press releases. However, this dataset is not publicly available due to the legal and licensing restrictions. Recently, Goldsack et al. (2022) constructed two open-access lay summarisation datasets from two academic journals (PLOS and eLife) in the biomedical domain. The datasets contain around 31k biomedical journal articles alongside expert-written lay summaries. Our dataset SCITECHNEWS is a valuable addition to the existing datasets, with coverage of diverse domains, including Computer Science, Machine Learning, Physics, and Engineering.

**Automatic Science Journalism.** Vadapalli et al. (2018) developed a pointer-generator network to generate blog titles from the scientific titles and their corresponding abstracts. Cao et al. (2020) built a manually annotated dataset for expertise style transfer in the medical domain and applied multiple style transfer and sentence simplification models to transform expert-level sentences into layman's language. The works most closely related to ours are Dangovski et al. (2021) and Goldsack et al. (2022). Both studies employed standard seq-to-seq models to generate news summaries for scientific articles. In our work, we propose a novel framework that integrates metadata information of scientific papers and scientific discourse structure to learn journalists' writing strategies.

## 3 The SCITECHNEWS Dataset

In this section, we introduce SCITECHNEWS, a new dataset for science journalism that consists of scientific papers paired with their corresponding press release snippets mined from ACM TechNews. We elaborate on how the dataset was collected and curated and analyze how it differs from other lay summarization benchmarks, both qualitatively and quantitatively.

### 3.1 Data Collection

ACM TechNews[3] is a news aggregator that provides regular news digests about scientific achievements and technology in the areas of Computer Science, Engineering, Astrophysics, Biology, and others. Digests are published three times a week as a collection of *press release snippets*, where each

---
[3] `https://technews.acm.org/`

snippet is written by a journalist and consists of a title, a summary of the press release, metadata about the writer (e.g., name, organization, date), and a link to the press release article.

We collect archived TechNews snippets between 1999 and 2021 and link them with their respective press release articles. Then, we parse each news article for links to the scientific article it reports about. We discard samples where we find more than one link to scientific articles in the press release. Finally, the scientific articles are retrieved in PDF format and processed using Grobid[4]. Following collection strategies of previous scientific summarization datasets (Cohan et al., 2018), section heading names are retrieved, and the article text is divided into sections. We also extract the title and all author names and affiliations.

Table 1 presents statistics of our dataset in comparison with datasets for lay, newswire, and scientific article summarization. Tokenization and sentence splitting was done using spaCy (Honnibal et al., 2020). In total, we gathered 29 069 press release summaries, from which 18 933 were linked to their corresponding press release articles. From these, 2431 instances –aligned rows in Table 1– were linked to their corresponding scientific articles. In this final subset, all instances have press release metadata (e.g., date of publication, author), press release summary and article, scientific article metadata (e.g., author names and affiliations), and scientific article body and abstract. We refer to this subset as SCITECHNEWS-ALIGNED, divide it into validation (1431) and test set (1000), and leave the rest of the unaligned data as non-parallel training data. Figure 6 in the appendix showcases a complete example of the aligned dataset. The train-test division was made according to the source and availability of each instance's corresponding scientific article, i.e., whether it is open access or not. The test set consists of only open-access scientific articles, whereas the validation set contains open-access as well as articles accessible only through institutional credentials. For this reason, we release the curated test set to the research community but instead provide download instructions for the validation set.

## 3.2 Dataset Analysis

We conduct an in-depth analysis of our dataset and report the knowledge domains covered and the

[4]https://github.com/kermitt2/grobid

| Dataset | Docs | Doc words | Summary words | sent. |
|---|---|---|---|---|
| **SciTechNews** | | | | |
| PR non-aligned | 29,069 | 612.56 | 205.93 | 6.74 |
| PR aligned | 2,431 | 780.53 | 176.07 | 5.72 |
| Sci. aligned | 2,431 | 7,570.27 | 216.77 | 7.88 |
| PLOS (Goldsack et al., 2022) | 27,525 | 5,366.70 | 175.60 | 7.80 |
| eLife (Goldsack et al., 2022) | 4,828 | 7,806.10 | 347.60 | 15.70 |
| LaySumm (Chandrasekaran et al., 2020) | 572 | 4,426.10 | 82.15 | 3.80 |
| Eureka-Alert (Zaman et al., 2020) | 5,204 | 5,027 | 635.60 | 24.3 |
| CNN / DailyMail (Hermann et al., 2015) | 311,971 | 685.12 | 51.99 | 3.78 |
| Newsroom (Grusky et al., 2018) | 1,210,207 | 770.09 | 30.36 | 1.43 |
| PubMed (Cohan et al., 2018) | 133,215 | 2,640.75 | 177.32 | 6.67 |
| arXiv (Cohan et al., 2018) | 215,913 | 5,282.27 | 237.79 | 8.87 |

Table 1: Comparison of aligned and non-aligned subsets of SCITECHNEWS with benchmark datasets for the tasks of lay, newswire, and scientific article summarization. For SCITECHNEWS, statistics for both the Press Release (PR) and the Scientific (Sci.) side are shown. The number of tokens (words) and sentences (sent.) are indicated as average.

| Source | #Instances Valid | Test |
|---|---|---|
| nature | 188 | 320 |
| arxiv | 263 | 231 |
| journals.aps | 21 | 73 |
| dl.acm | 67 | 64 |
| ieeexplore.ieee | 126 | 14 |
| usenix | 4 | 11 |
| journals.plos | 60 | 7 |
| author | 222 | 68 |
| other | 480 | 212 |
| Total | 1431 | 1000 |

Table 2: Most frequent sources of scientific articles in the validation and test set of SCITECHNEWS. The 'author' category refers to papers obtained from authors' personal websites.

variation in content type and writing style between scientific abstracts and press summaries.

**Knowledge Domain.** SCITECHNEWS gathers scientific articles from a diverse pool of knowledge domains, including Computer Science, Physics, Engineering, and Biomedical, as shown in Table 2. Sources include journals in Nature, ACM, APS, as well as conference-style articles from arXiv, IEEE, BioArxiv, among others. Note that a sizable chunk of articles was obtained from the authors' personal websites, as shown by the category 'author'.

**Readability.** The readability of scientific article abstracts and press summaries in our dataset is assessed using the following standard metrics: Flesch-Kincaid Grade Level (FKGL; (Kincaid et al., 1975)), Coleman-Liau Index (CLI; (Coleman and Liau, 1975)), Dale-Chall Readability Score (DCRS; (Dale and Chall, 1948)) and Gunning

| Metric | Sci | PR |
|---|---|---|
| **Readability** | | |
| FKGL↓ | 14.81 | 14.79 |
| CLI↓ | 15.17 | 14.23 |
| DCRS↓ | 11.08 | 11.13 |
| Gunning↓ | 16.33 | 16.75 |
| Average↓ | 14.35 | 14.23 |
| **Abstractivity (%)** | | |
| Novel unigrams↑ | 14.07 | 32.12 |
| Novel bigrams↑ | 47.32 | 72.50 |
| Novel trigrams↑ | 70.38 | 90.21 |

Table 3: Differences between scientific abstracts (Sci) and press release (PR) summaries in SCITECHNEWS, in terms of text readability (↓, the lower, the more readable) and percentage of novel ngrams – as a proxy for abstractiveness (↑, the higher, the more abstractive).

(Gunning, 1968).[5] These metrics aim to measure the simplicity or readability of a text by applying experimental formulas that consider the number of characters, words, and sentences in a text. For all these metrics, the lower the score, the more readable or simpler a text is. As shown in Table 3, the readability of abstracts and press summaries are on comparable levels (small gaps in scores), in line with observations in previous work in text simplification (Devaraj et al., 2021) and lay summarization (Goldsack et al., 2022). Nevertheless, all differences are statistically significant by means of the Wincoxin-Mann-Whitney test.

**Summarization Strategies.** We examined and quantified the differences in summarization strategies required in our dataset.

First, we assessed the degree of text overlap between the source document (i.e., the scientific article body) and either the abstract or the press summary as the reference summary, as shown in Figure 2. Specifically, we examine the extractiveness level of dataset samples in terms of extractive fragment coverage and density (Grusky et al., 2018).

When the reference summary is of non-scientific style (Fig. 2a), our dataset shows lower density than PLOS (Goldsack et al., 2022), a recent benchmark for lay summarization. This indicates that the task of science journalism, as exemplified by our dataset, requires following a less extractive strategy, i.e., shorter fragments are required to be copied verbatim from the source document. Similarly, when the reference summary is of scientific style (Fig. 2b), our dataset shows far lower density levels compared to ARXIV and a more concentrated

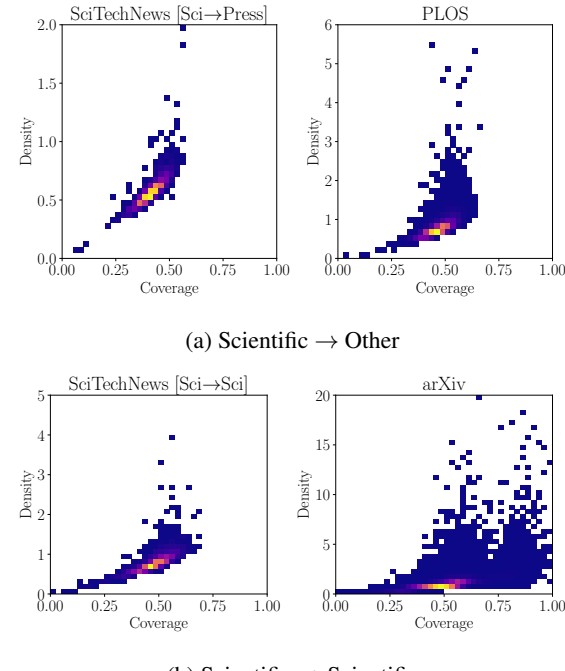

(a) Scientific → Other

(b) Scientific → Scientific

Figure 2: Extractivity levels of SCITECHNEWS and other summarization datasets in terms of coverage and density distributions, when the reference summary is of a different style (Other, a) and when it is of the same style (Scientific, b) as the source document. Warmer colors indicate more data entries.

distribution in terms of coverage. Such features indicate that SCITECHNEWS is much less extractive than ARXIV and constitutes a more challenging dataset for scientific article summarization, as we corroborated with preliminary experiments.

Second, we examined the amount of information in the reference summary not mentioned verbatim in the source document, a proxy for *abstractiveness*. Table 3 (second row) presents the percentage of novel n-grams in scientific abstracts (Sci) and press release summaries (PR). PR summaries show a significantly higher level of abstractivity than abstracts, indicating the heavy presence of information fusion and rephrasing strategies during summarization.

**Distribution of Named Entities.** What type of named entities are reported in a summary can be indicative of the writing style, more precisely of the intended audience and communicative goal of said summary. We quantify this difference by comparing the distribution of named entities[6] in scientific abstracts against that of press summaries.

As shown in Figure 3, press summaries show a

[5]Calculted using the textstat package.

[6]Extracted using the spaCy library.

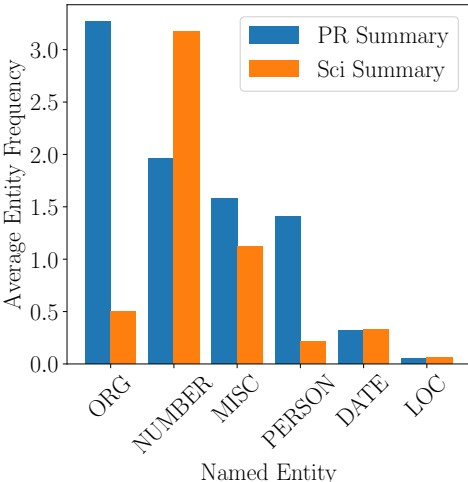

Figure 3: Average frequency of named entities in press release (PR) summaries and scientific abstracts (Sci).

high presence of organization and person entities, whereas scientific abstracts report mostly number entities. It is worth noting the low, however noticeable, presence of organization and person entities in the scientific abstracts. Upon closer inspection, these entities referred to scientific instruments and constants named after real-life scientists, e.g., *the Hubble telescope*. In contrast, person entities in press summaries most often referred to author names, whereas the organization entity referred to their affiliations.

**Discourse Structure**    Next, we examine the difference in scientific discourse structure between abstracts and press release summaries. We employ the model proposed by Li et al. (2021) trained on the PubMed-RCT dataset (Dernoncourt and Lee, 2017), and label each sentence in a summary with its rhetorical role, e.g., *background, conclusion, method*, among others[7]. Figure 4 presents the presence of rhetorical roles along with their relative positions in the summaries. Scientific abstracts tend to start with background information, then present methods, followed by results, and finish with conclusions. In contrast, press release summaries tend to emphasize conclusions way sooner than abstracts, taking the spotlight away from results and, to a lesser degree, from methods. Surprisingly, the relative presence of background information seems to be similar in both abstracts and press release summaries, in contrast with its emphasized presence in lay summaries, as reported in previous work (Goldsack et al., 2022).

---

[7]Li et al. (2021) report F-scores of 0.95 and 0.84 for scientific discourse tagging on two datasets from the biomedical

## 4    Problem Formulation and Modelling

We cast the problem of science journalism as an encoder-decoder generative task and propose a model that performs content planning and style transferring while summarizing the content. Given a scientific article text $D$, enriched with metadata information $M$, the task proceeds in two steps. First, a plan $s$ is generated conditioned on the input document, $p(s|D, M)$, followed by the summary $y$, conditioned on both the document and the plan, $p(y|s, D, M)$. Following Narayan et al. (2021), we train an encoder-decoder model that encodes an input document and learns to generate the concatenated plan and summary as a single sequence.

Let $D = \langle x_0, ..., x_N \rangle$ be a scientific article text, modeled as a sequence of sentences, let $M$ be the set of *author-affiliation* pairs associated with the said article, and let $Y$ be the target summary. We define $D' = \langle \mathrm{m}, m_0, .., m_{|M|}, t_0, x_0, .., t_N, x_N \rangle$ as the input to the encoder, where $\mathrm{m}$ is a special token indicating the beginning of metadata information, $m_i \in M$ is an author name concatenated to the corresponding affiliation, and $t_j$ is a label indicating the scientific rhetorical role of sentence $x_i$.

Given the encoder states, the decoder proceeds to generate plan $s$ conditioned on $D'$, $p(s|D'; \Theta)$, where $\Theta$ are the model parameters. The plan is defined as $s = \langle [\mathtt{PLAN}] s_0, ..., s_{|y|} \rangle$ where $s_k$ is a label indicating the rhetorical role sentence $y_k$ in summary $y$ should cover. Figure 1 shows an example of the annotated document and content plan. We use a Bart encoder-decoder architecture (Lewis et al., 2020) and train it with $D'$ as the source and $[s; y]$ (plan and summary concatenated) as the target. We call this model Bart$_{plan}$.

## 5    Experimental Setup

In this section, we elaborate on the baselines used and evaluation methods, both using automatic metrics and eliciting human judgments. Following previous work (Goldsack et al., 2022), we use the abstract followed by the introduction as the article body and prepend the metadata information as described in the previous section.

### 5.1    Comparison Systems

We compare against the following standard baselines: Extractive Oracle, obtained by greedily selecting $N$ sentences from the source document

---

and computational linguistic domains, respectively.

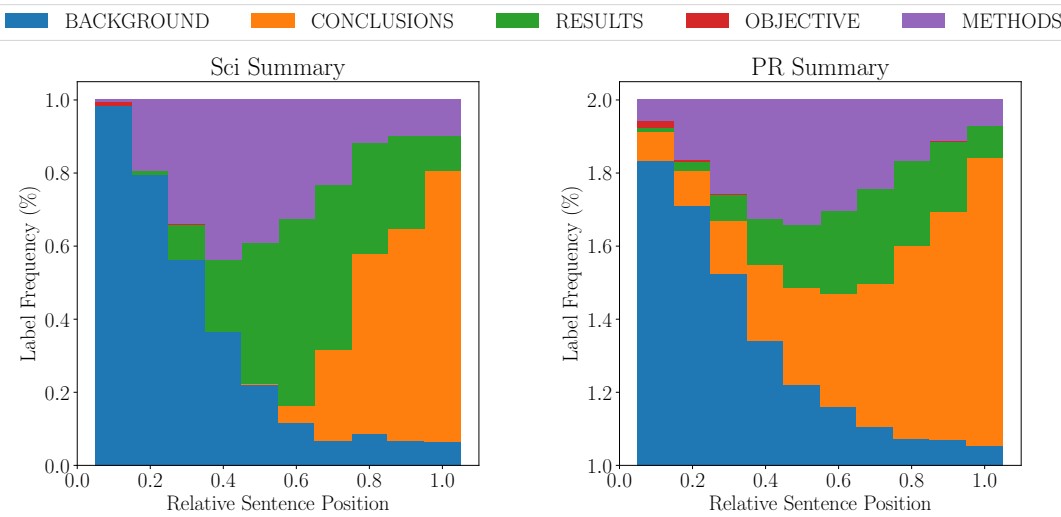

Figure 4: Distribution of scientific discourse tags in scientific abstracts (Sci) and press release (PR) summaries in SCITECHNEWS.

maximizing the ROUGE score (rouge 1 + rouge 2) against the reference summary; LEAD, which picks the first $N$ sentences of the source document; and RANDOM, which randomly selects $N$ sentences following a uniform distribution. For all our experiments, we use $N = 5$, the average number of sentences in PR summaries. Additionally, we report the performance of the scientific abstract, ABSTRACT, which provides an upper bound to extractive systems and systems that do not perform style transfer nor include metadata information.

For unsupervised baselines, we compare against LexRank (Erkan and Radev, 2004) and TextRank (Mihalcea and Tarau, 2004), two extractive systems that model the document as a graph of sentences and score them using node centrality measures. For supervised systems, we choose BART (Lewis et al., 2020) as our encoder-decoder architecture and use the pretrained checkpoints for BART-LARGE available at the HuggingFace library (Wolf et al., 2020). The following BART-based systems are compared: Bart$_{arx}$, finetuned on the ARXIV dataset (Cohan et al., 2018); Bart$_{SciT}$, finetuned on SCITECH-NEWS with only the abstract and introduction text as input, without metadata information or scientific rhetoric labels, and tasked to generate only the target summary without plan; and finally, Bart$_{meta}$, trained with metadata and article as input and summary without plan as the target.

Finally, we benchmark on recently proposed large language models (LLM) fine-tuned on the instruction-following paradigm: GPT-3.5-Turbo[8],

based on GPT3 (Brown et al., 2020); FlanT5-Large (Chung et al., 2022), fine-tuned on T5-3B (Raffel et al., 2020); and Alpaca 7B (Taori et al., 2023), an instruction-finetuned version of LLaMA (Touvron et al., 2023). We employ the same instruction prompt followed by the abstract and introduction for all systems, *"Write a report of this paper in journalistic style."*

## 5.2 Evaluation Measures

Given the nature of our task, we evaluate the intrinsic performance of our models across the axes of summarization and style transfer.

**Summarization.** The informativeness, relevance, and fluency of the generated summaries are evaluated using ROUGE 1, 2, and L, respectively (Lin, 2004).[9] Semantic relevance is evaluated with BertScore (Zhang et al.) using RoBERTa-large as base model (Liu et al., 2019) and in-domain importance weighting.[10] All evaluations were made over the summary text after stripping away the generated content plan.

**Style Transfer** To distinguish between press release style and scientific style, we train a binary sentence classifier using press release summary sentences from the unaligned split of SCITECH-NEWS as positive samples, and an equal amount of sentences from scientific abstracts from arXiv (Cohan et al., 2018) as negative examples. We

---

[8]We used model `gpt-3.5-turbo-0301` in `https://platform.openai.com/docs/models`.

[9]As calculated by the `rouge-score` library.

[10]BertScore has been proven a reliable metric when equipped with importance weighting in highly technical domains such as medical texts (Miura et al., 2021; Hossain et al., 2020).

use the RoBERTa-base model as implemented in the huggingface library in a sequence classification setup. Then, the style score $sty(S)$ of summary $S$ is defined as the probability of the positive class, averaged over all sentences in $S$.

**Faithfulness.** Factual consistency of generated summaries with respect to their source document is quantified using QuestEval (Scialom et al., 2021).

**Human Evaluation.** We take a random sample of 30 items from the test set and conduct a study using *Best-Worst Scaling* (Louviere et al., 2015), a method that measures the maximum difference between options and has been shown to produce more robust results than rating scales (Kiritchenko and Mohammad, 2017). Human subjects were shown the source document (abstract, introduction, and metadata) along with the output of three systems. They were asked to choose the *best* and the *worst* according to the following dimensions: (1) *Informativeness* – how well the summary captures important information from the document; (2) *Factuality* – whether named entities were supported by the source document;[11] (3) *Non-Redundancy* – if the summary presents less repeated information; (4) *Readability* – if the summary is written in simple terms; (5) *Style* – whether the summary text follows a journalistic writing style; and finally, (6) *Usefulness* – how useful would the summary be as a first draft when writing a press release summary of a scientific article. Systems are ranked across a dimension by assigning them a score between $-1.0$ and $1.0$, calculated as the difference between the proportion of times it was selected as *best* and selected as *worst*. See Appendix D for more details.

## 6 Results and Discussion

In this section, we present and discuss the results of our automatic and human evaluations, provide a comprehensive analysis of the factuality errors our systems incur, and finish with a demonstration of controlled generation with custom user plans.

### 6.1 Automatic Metrics

**Informativeness and Fluency.** Table 4 presents the performance of the compared systems in terms of ROUGE and BertScore. We notice that the extractive upper-bounds, ABSTRACT and EXT-ORACLE, obtain relatively lower scores compared

---

[11]We consider names of people, locations, organization, as well as numbers.

| Systems | R1 | R2 | RL | BSc |
|---|---|---|---|---|
| ABSTRACT | 32.94 | 6.26 | 28.84 | 81.20 |
| EXT-ORACLE | 39.73 | 10.43 | 34.10 | 84.49 |
| Lead | 32.46 | 5.79 | 28.17 | 83.81 |
| Random | 29.58 | 3.99 | 25.50 | 82.60 |
| LexRank | 31.40 | 5.21 | 27.16 | 82.98 |
| TextRank | 31.86 | 5.38 | 27.38 | 82.92 |
| $\text{Bart}_{arx}$ | 32.28 | 6.01 | 28.12 | 82.81 |
| $\text{Bart}_{SciT}$ | 36.42 | 7.51 | 31.71 | 84.12 |
| $\text{Bart}_{meta}$ | 38.07 | **9.03** | 33.14 | 84.76 |
| $\text{Bart}_{plan}$ | **38.84*** | 8.89 | **33.50*** | **84.78** |
| Alpaca | 21.24 | 3.24 | 18.16 | 81.20 |
| FlanT5-large | 26.26 | 4.98 | 20.13 | 80.98 |
| GPT-3.5-Turbo | 35.67 | 6.75 | 28.68 | 82.86 |

Table 4: Results in terms of ROUGE-F1 scores (R1, R2, and RL) and BertScore F1 (BSc). Best systems in **bold**. *: statistically significant w.r.t. to the closest baseline with a 95% bootstrap confidence interval.

| Systems | CLI↓ | QEval↑ | Sty↑ |
|---|---|---|---|
| $\text{Bart}_{arx}$ | 15.33 | **47.90** | 0.18 |
| $\text{Bart}_{SciT}$ | 13.70 | 36.54 | 0.96 |
| $\text{Bart}_{meta}$ | **13.43** | 36.91 | **0.98** |
| $\text{Bart}_{plan}$ | 13.55 | 38.16 | **0.98** |
| Alpaca | 13.82 | 38.00 | 0.25 |
| FlanT5-large | 16.36 | 44.36 | 0.10 |
| GPT-3.5-Turbo | 16.36 | 46.51 | 0.81 |
| PR Summary | 16.52 | 33.95 | 0.99 |

Table 5: Performance of systems in terms of readability (CLI), faithfulness (QuestEval score; QEval), and our style score (Sty). (↑,↓): higher, lower is better.

to previously reported extractive upper-bounds in lay summarization (Goldsack et al., 2022). This further confirms that the kind of content covered in press release summaries and scientific abstracts are fundamentally different, as explored in Section 3.2. For the abstractive systems, we notice that $\text{Bart}_{meta}$ significantly improves over $\text{Bart}_{SciT}$, highlighting the critical importance of adding metadata information to the source document. Generating a scientific rhetorical plan as part of the output further improves informativeness (Rouge-1) and fluency (Rouge-L), as well as semantic relevance (BertScore) of the produced content. It is worth noting that the assessed LLMs, Alpaca, FlanT5, and GPT-3.5, underperform the proposed models, indicating that the task poses a significant challenge to them under zero-shot conditions.

**Readability, Faithfulness, and Style.** First, we find that adding metadata and plan information reduces syntactic and lexical complexity and improves faithfulness, as shown in Table 5. Inter-

| System | Inf. | N-Rd. | Fact. | Read. | Sty. | Use. |
|---|---|---|---|---|---|---|
| Bart$_{meta}$ | 0.13 | -0.31 | -0.33 | 0.01 | 0.16 | -0.22 |
| Bart$_{plan}$ | **0.08** | **0.08** | -0.10 | **0.22** | **0.30** | **0.02** |
| GPT-3.5 | -0.07 | -0.01 | **0.02** | -0.23 | -0.24 | -0.21 |
| PR Sum. | 0.58 | 0.68 | 0.43 | 0.79 | 0.91 | 0.57 |

Table 6: System ranking according to human judgments, along (Inf)ormativeness, (Non-Red)undancy, (Fact)uality, (Read)ability, Press Release (Sty)le, and (Use)fulness. Best system is shown in **bold**.

estingly, FlanT5 and GPT-3.5 generate seemingly more complex terms, followed by Bart$_{arx}$. Upon inspection, these systems showed highly extractive behavior, i.e., the produced summaries are mainly composed of chunks copied from the input *verbatim*. We hypothesize that this property also inflated their respective faithfulness scores. Note that gold PR summaries show a low QEval score, indicating that faithfulness metrics based on pre-trained models are less reliable when the source and target texts belong to a highly technical domain or differ in writing style. In terms of style scoring, we observe that models finetuned on our dataset are capable of producing summaries in press release style, a specific kind of newswire writing style. See Appendix E for a few generation examples by Bart$_{meta}$ and Bart$_{plan}$.

## 6.2 Human evaluation

Table 6 shows the results of our human evaluation study, comparing models effective at encoding metadata (Bart$_{meta}$), generating a plan (Bart$_{plan}$) and a strong LLM baseline (GPT-3.5). Inter-annotator agreement – Krippendorff's alpha (Krippendorff, 2007) – was found to be 0.57. Pairwise statistical significance was tested using a one-way ANOVA with posthoc Tukey-HSD tests and 95% confidence interval. The difference between preferences across dimensions was found to be significant ($p < 0.01$) for the following pairs: expert-written gold PR summaries vs. all systems, in all dimensions; for Non-Redundancy, Bart$_{plan}$ and GPT3.5 against Bart$_{meta}$; for Factuality, Bart$_{meta}$ vs GPT-3.5; for Readability, Bart$_{plan}$ vs all systems and Bart$_{meta}$ vs GPT-3.5; and for Style and Usefulness, all pairs combinations were significant.

The results indicate the following. First, scores for PR summaries are higher than machine-generated text, further confirming the difficulty of the task and showing ample room for improvement. Second, Bart$_{meta}$'s rather low scores in Non-

Redundancy and Style can be due to its memorization of highly frequent patterns in the dataset, e.g., *'researchers at the university of ...'*. In contrast, Bart$_{plan}$ generates more diverse and stylish text. Third, whereas GPT-3.5's high Factuality score can be attributed to the difference in the number of architecture parameters, its low Readability and Style scores indicate that the simplification and stylization of complex knowledge still pose a significant challenge. Finally, in terms of Usefulness, users preferred Bart$_{plan}$ as a starting draft for writing a press release summary, demonstrating the model's effectiveness for this task.

## 6.3 Factuality Error Analysis

We further analyzed the types of factuality errors our systems incurred on. We uniformly sampled 30 instances from the test set and manually annotated their respective reference summary and summaries generated by Bart$_{plan}$ and GPT3.5-Turbo.

We adapt the error taxonomy employed in Goyal and Durrett (2021) and consider three categories at the span level:[12] (i) *Entity-related*, when the span is a named entity (same entity categories considered in Section 3.2.); (ii) *Noun Phrase-related*, when the span is an NP modifier; and (iii) *Other Errors*, such as repetitions and grammatical errors. Each error category except 'Other' is further divided into sub-categories: Intrinsic, Extrinsic, and World Knowledge, depending on where the supporting information is found (Cao and Wang, 2021). Intrinsic errors are caused when phrases or entities found in the input are generated in the wrong place. In contrast, extrinsic errors happen when the generated span cannot be supported by the input or any external source (e.g., Wikipedia). Finally, word knowledge errors are caused when the span cannot be verified with the input but it can be verified using external knowledge, e.g. author X being the director of the C.S. department at university Y.

Table 7 presents the proportion of error categories found in the inspected summaries, along with the total number of error spans found for each system. It is worth noting that the total number of errors follows the ranking trend in Table 6, with PR summaries having the least number of errors, followed by GPT3.5, and then Bart$_{plan}$. First, we observe that reference summaries exhibit only Entity and NP-related errors of type World Knowledge.

---

[12]The event-related category is not considered here since the source documents in SCITECHNEWS do not contain events.

| | **Oracle Plan & Generated Summary** |
|---|---|

**[PLAN]** [AUTHOR] [CONCLUSIONS] I [CONCLUSIONS] I [AUTHOR] I [CONCLUSIONS] **[SUMMARY]**
Researchers at the University of Basel in Switzerland say they have developed a method for generating a secure quantum key via photonic conversion. The method adds artificial noise to the initial measurement data, which cannot be known or controlled by an adversary, to guarantee that the key will be obtained by both Alice and Bob, who are connected by a quantum channel. By adding the noise, the researchers were able to increase the threshold at which they can guarantee the key is obtained. Although the researchers say their method is not perfect, they believe it is still effective and could lead to a new generation of highly secure and provable quantum key distribution. They note that the new method provides security even when assumptions about the security of QKD are not fulfilled.

**User Plan & Generated Summary**

**[PLAN]** [AUTHOR] I [BACKGROUND] I [METHODS] I [RESULTS] [CONCLUSIONS] **[SUMMARY]**
Researchers at the University of Basel in Switzerland say they have developed a method for generating a secure quantum key via photonic conversion. The method involves adding artificial noise to the initial measurement data, which cannot be known or controlled by an adversary, to increase the probability that the key is obtained. Although the researchers say the method is not fully device-independent, they say it provides security even when assumptions are made about the security of the quantum key distribution. "There is a fundamental obstacle in the development of QKD, i.e.,, the requirement that an adversary cannot fully control the quantum channel," says Basel Professor Sangouard Bancal. However, he says the method provides sufficient bounds on the minimum required global detection efficiency to ensure that the information sent over the qubit channel is good and accurate.

Figure 5: Example of generation by Bart$_{plan}$ conditioned to a user plan. Text and plan labels are color-coded.

| System | Entity | | | Noun Phrase | | | Other | Total |
|---|---|---|---|---|---|---|---|---|
| | Int. | Ext. | W.K. | Int. | Ext. | W.K. | | |
| PR Sum. | 0.0 | 0 | 0.79 | 0.0 | 0.0 | 0.21 | 0 | 43 |
| Bart$_{plan}$ | 0.1 | 0.34 | 0.20 | 0.07 | 0.16 | 0.02 | 0.11 | 61 |
| GPT3.5 | 0.0 | 0.08 | 0.0 | 0.02 | 0.18 | 0.0 | 0.72 | 50 |

Table 7: Proportion of factuality errors in different systems with error proportions normalized by system.

The majority of them include completed names of affiliated institutions (e.g., the metadata mentioning only the abbreviation 'MIT' but the reference summary showing the complete name), country names where these institutions are located, or the position an author holds within an institution. We also found cases where an author held more than one affiliation, with only one of these being listed in the metadata and another mentioned in the reference summary. Second, we observe that Bart$_{plan}$ extrinsically hallucinates mostly entities (e.g., country names, 34% of all its errors), followed by extrinsic NPs. Among intrinsic errors, entity-related ones included mixed-ups of author first names, last names, and affiliations, whilst NP-related errors included mistaking resources mentioned in the source (e.g. a github repository) for institutions. In contrast, GPT3.5 produced a sizable amount of extrinsic hallucinations of noun phrases, most of them stating publication venues (e.g., '*In a paper published in ...*'). Since the metadata added to the source does not include publication venue, the model must have surely been exposed to this kind of information during training. However, somewhat surprisingly, GPT3.5-Turbo did not exhibit world knowledge errors of any kind, potentially highlighting the conservative approach to generation followed during its training. Errors of type 'Other' consisted mainly of orphaned phrases at the beginning of generation, i.e., the model starts the generation by attempting

to continue the last sentence of the input in a 'continue the story' fashion. We hypothesize that the GPT3.5 model employed (checkpointed at March, 2023) struggled with the length of the input, reaching a point where the prompt instruction (stated at the beginning of the input) is completely ignored and the model just continues the 'story' given.

## 6.4 Controlled Generation with User Plans

The proposed framework opens the possibility of controlling the content and the rhetorical structure of the generated summary by means of specifying custom plans of rhetorical labels. Figure 5 presents an example of this, showing that Bart$_{plan}$ generates content from all the requested roles in the plan, following most of the precedence order stated.

## 7 Conclusions

This paper presents a novel dataset, SCITECH-NEWS, for automatic science journalism. We also propose a novel approach that learns journalistic writing strategy and style by leveraging the paper's discourse structures. Through extensive automatic evaluation and human evaluation with baseline methods (e.g., ChatGPT and Alpaca), we find that our models can generate high-quality informative news summaries that closely resemble those crafted by professional journalists.

## 8 Limitations

The introduced dataset is in English, as a result, our models and results are limited to English only. Future work can focus on the creation of datasets and the adaptation of science journalism to other languages. Also of relevance is the limited size of our dataset, and the potential lack of balanced coverage on the reported knowledge domains. Finally, LLMs results suggest that a more extensive prompt engineering might be critical to induce generation with adequate press release journalistic style.

Another limitation of our approach is the usage of only author and affiliation metadata as additional input information. We decide to only consider this metadata for the following reason. Considering the distribution of named entities found in Press Release reference summaries (analyzed in Section 3.2 and depicted in Figure 3), it is worth noting that entities of type Organization and Person are the most frequent entities – after numbers and miscellaneous. Hence, we decided to restrict the usage of metadata in our framework to author's names and affiliations. However, other metadata information was collected, both from the scientific article (e.g. publication venue and year, title) and press release articles (e.g. title, PR publication date, journalistic organization), as detailed in Section 3.1. We include the complete metadata in the released dataset so that future investigations can leverage them.

## 9 Ethics Statements

The task of automatic science journalism is intended to support journalists or the researchers themselves in writing high-quality journalistic content more efficiently and coping with information overload. For instance, a journalist could use the summaries generated by our systems as an initial draft and edit it for factual inconsistencies and add context if needed. Although we do not foresee the negative societal impact of the task or the accompanying data itself, we point at the general challenges related to factuality and bias in machine-generated texts, and call the potential users and developers of science journalism applications to exert caution and follow up-to-date ethical policies.

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

# A Example Snippet

Figure 6 showcases a complete example of an ACM TechNews snippet paired with the scientific paper it talks about.

# B Training and Resource Details

BART models were trained on two NVIDA A100 GPUs, each with 80GB of memory, using Adam optimizer (Loshchilov and Hutter, 2018) with a learning rate of $1e-6$, batch size of $128$, for a maximum of $5.000$ steps. LLM experiments were run on one NVIDIA A100 40G graphic card. For FlatT5-Large, we use a maximum length of 256, beam size of 5, temperature of $0.9$, top_k of 100, and use early stopping. For Alpaca, the default generation parameters are used.

# C Supplementary Readability and Faithfulness Evaluation

Table 8 presents supplementary performance results of our systems w.r.t. readability and faithfulness. In addition to QuestEval, we report entailment-based scores SummaC (Laban et al., 2022) and Adversarial NLI (Nie et al., 2020).

# D Human Evaluation

Following a typical human evaluation setup as in the previous literature (Yao et al., 2022), we recruited 5 volunteers for human evaluation, all PhD students in Computer Science, and hosted the study on an internal server. Participants were selected so that their area of expertise do not overlap significantly with the topic of the articles in the study.

| | |
|---|---|
| **ACM TechNews Snippet** | |

**Title**
Researchers Say They've Found a Wildly Successful Bypass for Face Recognition Tech
**Press Summary**
Computer scientists at Israel's Tel Aviv University (TAU) say they have developed a "master face" method for circumventing a large number of facial recognition systems, by applying artificial intelligence to generate a facial template. The researchers say the technique exploits such systems' usage of broad sets of markers to identify specific people; producing facial templates that match many such markers essentially creates an omni-face that can bypass numerous safeguards. ...
**Press Release**
In addition to helping police arrest the wrong person or monitor how often you visit the Gap, facial recognition is increasingly used by companies as a routine security procedure: it's a way to unlock your phone or log into social media, for example. This practice comes with an exchange of privacy for the promise of comfort and security but, according to a recent study, ...

**Scientific Article**

**Title**
Generating Master Faces for Dictionary Attacks with a Network-Assisted Latent Space Evolution
**Abstract**
A master face is a face image that passes face-based identity-authentication for a large portion of the population. These faces can be used to impersonate, with a high probability of success, any user, without having access to any user-information. We optimize these faces, by using an evolutionary algorithm in the latent embedding space of the StyleGAN face generator. Multiple evolutionary strategies are compared, ...
**Main Body**
I. INTRODUCTION
In dictionary attacks, one attempt to pass an authentication system by sequentially trying multiple inputs. In real-world biometric systems, one can typically attempt only a handful of inputs before being blocked. However, the matching in biometrics is not exact, and the space of biometric data is not uniformly distributed. This may suggest that with a handful of samples, one can cover a larger portion of the population. ...

Figure 6: Example from our SCITECHNEWS dataset showing a complete scientific article (title, abstract, and main body; bottom) and its associated ACM TechNews snippet (title, press summary, and press release article; top).

The study comprised a sample of 30 scientific articles, and each participant annotated all articles but were allowed to do so in their own pace and time. Moreover, we discouraged participants from doing more than 5 articles in a single sitting.

As shown in Figure 7, participants were shown a brief description of the task, followed by the scientific article (abstract and introduction), metadata information, along with the output of three systems (Narayan et al., 2019). Then, they were asked to select the best and worst systems according to the dimensions mentioned in Section 4. In case there was no significant difference between all systems, participants were instructed to select all systems as best and worst. Similarly, if there was no significant difference between the best and second best, or worst and second worst, participants were allowed to select both systems. The score of a system is calculated as the proportion of times it was selected as *best* minus the proportion of times it was selected as *worst*. Hence, the score can be a value between -1 and 1.

## E Example of System Outputs

Figure 8 and 9 showcase press release summaries from SCITECHNEWS and the corresponding summaries generated by systems $Bart_{meta}$ and $Bart_{plan}$.

## F Controlling Generation with User Plans

Figure 10 presents a complete example of summary generation with Oracle, system-generated, and user plans. Notice that $Bart_{plan}$ generates content from all the requested roles in the plan, following most of the precedence order stated.

| System | Readability | | | | Faithfulness | | |
|---|---|---|---|---|---|---|---|
| | FKGL↓ | CLI↓ | DCRS↓ | Gunning↓ | QEval↑ | Sumc↑ | ANLI↑ |
| Bart$_{arx}$ | 15.21 | 15.33 | 11.71 | 17.27 | **47.90** | **80.12** | **69.95** |
| Bart$_{SciT}$ | 15.40 | 13.70 | 10.74 | 17.36 | 36.54 | 28.62 | 22.77 |
| Bart$_{meta}$ | 15.22 | **13.43** | **10.66** | 17.21 | 36.91 | 28.33 | 25.30 |
| Bart$_{plan}$ | 15.35 | 13.55 | 11.03 | 17.59 | 38.16 | 28.54 | 28.96 |
| Alpaca | **12.21** | 13.82 | 11.00 | **14.04** | 38.00 | 67.97 | 48.76 |
| FlanT5-large | 15.12 | 16.36 | 11.92 | 16.97 | 44.36 | 73.76 | 63.26 |
| GPT-3.5-Turbo | 14.68 | 16.52 | 11.29 | 16.03 | 46.51 | 55.02 | 49.82 |
| PR Summary | 15.16 | 14.61 | 11.51 | 17.25 | 33.95 | 27.09 | 31.10 |

Table 8: Supplementary performance results of systems in terms of readability (the lower the better) and faithfulness (the higher the better). QEval: QuestEval; Sumc: SummaC; ANLI: Adversarial NLI.

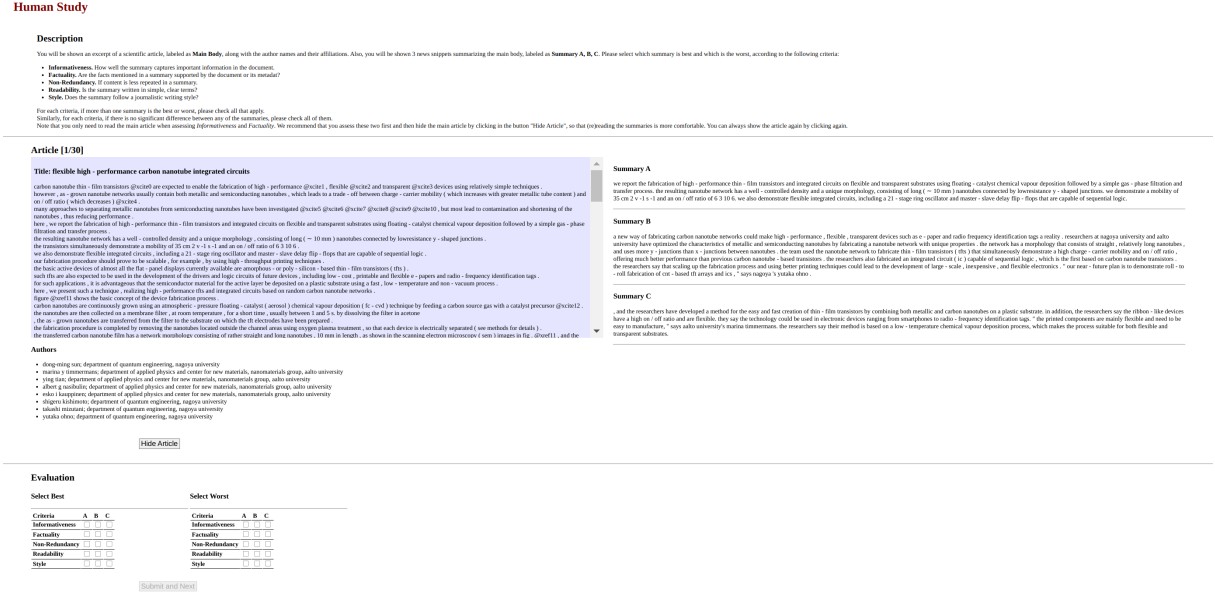

Figure 7: Web interface used for the human study introduction in Section 4.

**Press Release Summary**

university of illinois at urbana - champaign ( uiuc ) scientists have created a computer model of a simple brain network based on that of a sea slug , which was trained to find food . the asimov simulated slug was placed in a confined setting to randomly encounter food pellets that were alternately delicious or noxious , emitting a characteristic odor to either attract or repulse the slug . asimov was programmed to experience satiation as well as reward , and the addition of a rewarding but nutritionally empty pellet enticed the slug to pursue it to the exclusion of all else . although consuming this pellet caused satiation , that sense of fullness dissipated over time , with a homeostatic plasticity mechanism causing asimov to experience withdrawal . uiuc 's rhanor gillette said , " by watching how this brain makes sense of its environment , we expect to learn more about how real - world brains work . "

**Bart$_{meta}$**

researchers at the university of illinois at urbana - champaign ( uiuc ) have developed an artificial intelligence ( ai ) system that simulates the addictive process, incorporating a sense of taste and disgust as well as learning mechanisms that drive addiction. the team used a virtual reality to create cyberslug, an agent that forages based on value and probability, with the artificial intelligence system learning to approach or avoid prey based on information such as nutritional value, comfort, and the potential of infection. asimov was exposed to stimuli that it did not like, such as choking, and would eventually feel the effects of withdrawal. uiuc's rhanor gillette said, " through that simulation, we can see that ai has a very similar aesthetic sense to addiction and that the feelings that people with addiction have may be similar to those that other people have. "

**Bart$_{plan}$**

a university of illinois at urbana - champaign ( uiuc ) scientist has developed an artificial intelligence ( ai ) system that simulates a drug - like addiction, with the goal of making it easy to visualize the effects of such addiction on a computer. the program gives the ai visual cues to expect to receive a high reward, but also warns the agent of the potential unpleasant feelings that may follow after consuming that reward. ekaterina gribkova's lab developed the artificial intelligence to replicate the neural circuitry involved in addiction, and showed that ai can be programmed to watch movies, listen to music, and act on visual cues. said gribkov, " if you want to understand what addiction is like, you can start with the analogs and find out how the addictive processes are connected to each other. "

**GPT-3.5**

Aesthetic decision-making in primitive generalist foragers, shedding light on the coevolutionary origins of aesthetics and addiction. According to a recent study published in the journal PLOS Biology, a rudimentary aesthetic sense can be found in the stimulus valuations and cost-benefit decisions made by primitive generalist foragers. This aesthetic sense is based on factors governing personal economic decisions such as incentive, appetite, and learning. The study found that the addictive process is an extreme expression of aesthetic dynamics. The researchers used an interactive, agent-based model called Asimov to reproduce a simple aesthetic sense from known neural relations of cost-benefit decision-making in foraging. In the presence of very high reward, an addiction-like process emerges. A drug-like prey provides extreme reward with no nutritive value, initiating high selectivity and prolonged cravings for the drug through reward learning.

Figure 8: First example of reference summary along with corresponding output summaries of systems Bart$_{meta}$, Bart$_{plan}$, and GPT-3.5.

**Press Release Summary**

a team of researchers from the massachusetts institute of technology ( mit ) , arizona state university , and the university of massachusetts at amherst developed a new control system that enhances the speed and agility of legged robots as they leap across gaps . the control system algorithmically processes and translates real - time forefront video input into instructions for bodily movement . the researchers combined the best elements of controllers that do not incorporate vision into a separate module that handles vision in real time , and trained the controller through reinforcement learning . tests of the system when installed into mit 's mini cheetah robot found it outperformed other systems that use a single controller , enabling it to successfully cross 90 % of physical terrains .

**Bart$_{meta}$**

researchers at the massachusetts institute of technology ( mit ) have developed a quadruped robot that can successfully jump through wide gaps, and across uneven terrain. the team notes this is an important milestone toward the development of " blind walking, " which relies on having the robot navigate without vision. " without vision, the robot would either step in the gap or fall or otherwise treat the gap as an obstacle and stop, " the researchers say. their approach relies on making the robot anticipate how the ground will behave next. this enables the robot to execute novel maneuvers, such as jumping over large gaps that require planning beforehand. in addition, the team uses a control scheme called depth - based impulse control ( dic ), which provides the robot with a flexible way to control the system when it is near an unseen obstacle. dic involves the robot acting on impulse based on small changes in the surrounding environment, which can be very useful when controlling a robot that has no visual information. said mit's pulkit agrawal, " we're trying to create a system that can be adaptable to any type of environment, even though it has visual impairment. "

**Bart$_{plan}$**

themasses institute of technology's tao chen has designed a quadruped robot that can successfully jump across small gaps, but would not know if it is safe to continue moving forward.
the robot uses a depth - based impulse control system, which avoids the traditional control scheme of arranging the robot in a straight line along the edge of the obstacle, but instead jumps depending on the momentum of the body.
as the robot approaches the edge, it senses an obstacle getting closer, and then uses light to predict how far it is from the robot and whether it will be able to jump across the gap.
to control the robot, chen says, " we need to provide a large range of possible control schemes that the robot can use to produce dynamic behavior, such as jumping, using reinforcement learning. "

**GPT-3.5**

A team of researchers has developed a new method for creating agile locomotion in robotic quadrupeds that can navigate discontinuous terrains, such as those with gaps or obstacles. The method, called depth-based impulse control (DIC), uses visual inputs to plan ahead and execute agile behaviors beyond robust walking, such as jumps. However, dynamic motion can result in significant motion of onboard sensors, which poses new challenges for real-time visual processing. DIC combines model-free learning with explicit model-based optimization of ground reaction forces to regularize behavior. The team evaluated the method in both simulation and the real world, using a gap-world environment containing flat regions and randomly placed variable-width gaps. The ability to traverse discontinuous terrains with agility and terrain awareness reinforces the need for robust control, especially as the robotics community seeks to construct legged systems that can navigate novel and complex landscapes.

Figure 9: Second example of reference summary along with corresponding output summaries of systems Bart$_{meta}$, Bart$_{plan}$, and GPT-3.5.

**Oracle Plan & Press Release Summary**
[PLAN] [AUTHOR] [CONCLUSIONS] | [CONCLUSIONS] | [AUTHOR] | [CONCLUSIONS] **[SUMMARY]**
Researchers at Switzerland's University of Basel and ETH Zurich have established the theoretical foundations for 100% secure communication, ensuring protection not just against quantum computers but also in instances where the communication devices' operational mechanisms and trustworthiness are unknown. The protocol adds artificial noise to information about the cryptographic key, guaranteeing eavesdroppers receive so little actual data as to render the protocol unbreakable. Nicolas Sangouard from the University of Basel said, "Since the first small-scale quantum computers are now available, we urgently need new solutions for protecting privacy. Our work represents a significant step toward the next milestone in secure communications."

**Oracle Plan & Generated Summary**
[PLAN] [AUTHOR] [CONCLUSIONS] | [CONCLUSIONS] | [AUTHOR] | [CONCLUSIONS] **[SUMMARY]**
Researchers at the University of Basel in Switzerland say they have developed a method for generating a secure quantum key via photonic conversion. The method adds artificial noise to the initial measurement data, which cannot be known or controlled by an adversary, to guarantee that the key will be obtained by both Alice and Bob, who are connected by a quantum channel. By adding the noise, the researchers were able to increase the threshold at which they can guarantee the key is obtained. Although the researchers say their method is not perfect, they believe it is still effective and could lead to a new generation of highly secure and provable quantum key distribution. They note that the new method provides security even when assumptions about the security of QKD are not fulfilled.

**Generated Plan & Generated Summary**
[PLAN] [AUTHOR] [BACKGROUND] | [BACKGROUND] [CONCLUSIONS] | [CONCLUSIONS] [METH-ODS] | [RESULTS] | [METHODS] [RESULTS] **[SUMMARY]**
Researchers at the University of Basel in Germany have developed a method for generating a key for a quantum computer in which both Alice and Bob are connected by a quantum channel. The method adds artificial noise to the initial measurement data to increase the probability that the signals sent over the quantum channel are successfully received. However, in practice, only a small amount of information about the key can be obtained. In a fully device-independent QKD, the researchers assume that any information contained in the measurement device must be known and controlled by an external party. Unlike traditional quantum key distribution, Eve does not directly control the measurement outcome, and in the absence of such information, the key is sent to an untrusted source who also distributes the information to the other user. By adding artificial noise, Eve can not ensure that the information in the raw key is real. "Our method provides a way to significantly relax the threshold in such a way as to provide provable security," says Basel Professor Sangouard.

**User Plan & Generated Summary**
[PLAN] [AUTHOR] | [BACKGROUND] | [METHODS] | [RESULTS] [CONCLUSIONS] **[SUMMARY]**
Researchers at the University of Basel in Switzerland say they have developed a method for generating a secure quantum key via photonic conversion. The method involves adding artificial noise to the initial measurement data, which cannot be known or controlled by an adversary, to increase the probability that the key is obtained. Although the researchers say the method is not fully device-independent, they say it provides security even when assumptions are made about the security of the quantum key distribution. "There is a fundamental obstacle in the development of QKD, i.e.,, the requirement that an adversary cannot fully control the quantum channel," says Basel Professor Sangouard Bancal. However, he says the method provides sufficient bounds on the minimum required global detection efficiency to ensure that the information sent over the qubit channel is good and accurate.

Figure 10: An example of generating summaries with custom user content plans and Bart$_{plan}$. Text and corresponding rhetorical roles and color-coded.