# OpenReview forum: "`Don't Get Too Technical with Me': A Discourse Structure-Based Framework for Automatic Science Journalism"
_EMNLP/2023/Conference — EMNLP 2023 Main_

### Official Review · Reviewer_MWo1 · 2023-08-04

**Soundness:** 3

**Excitement:**

3: Ambivalent: It has merits (e.g., it reports state-of-the-art results, the idea is nice), but there are key weaknesses (e.g., it describes incremental work), and it can significantly benefit from another round of revision. However, I won't object to accepting it if my co-reviewers champion it.

**Paper Topic And Main Contributions:**

This paper studies the task of summarizing scientific abstracts into scientific news for public audience. It presents a public dataset which contains more domains than existing datasets. The authors propose methods to utilize the discourse structure in the input abstract and show improvement.

**Reasons To Accept:**


* Constructed a new dataset for this task which contains more domains than existing datasets. The new dataset could be beneficial for the research community.

* Presented some comprehensive analysis in Section 3 to show the style difference between scientific abstracts and press summaries. The analysis could be helpful and inspire future work to leverage these characteristics.

* Proposed to utilize discourse structure as extra information, which was shown to improve the model.

* The paper is well-written and easy to follow.


**Reasons To Reject:**


* The contribution of using meta information is quite weak. For the meta information, I originally thought it was to use a set of meta data to help summarize texts.  However in the paper it is basically using the author name and affiliation to fill in necessary information.

* The motivation of the two-step framework is unclear, so more analyses about this are needed. Specifically, why is it useful to generate the plan first before generating the summary? It is not clear if the improvement of BART_plan mostly comes from the discourse role labels added in the inputs.

**Reproducibility:**

4: Could mostly reproduce the results, but there may be some variation because of sample variance or minor variations in their interpretation of the protocol or method.

**Reviewer Confidence:**

3: Pretty sure, but there's a chance I missed something. Although I have a good feel for this area in general, I did not carefully check the paper's details, e.g., the math, experimental design, or novelty.

---

> ### Author Rebuttal · Authors · 2023-08-28
>
> Thank you for the feedback and comments. We appreciate the attention to the points raised and provide a thorough discussion which will be integrated into the final version.
>
> ## Contribution of extra signal in the source and target
>
> The comparison systems introduced in Section 5.1. intend to present an incremental story of how metadata and rhetorical labels in the source improve summary qualities as well as the benefits of generating a content plan before the actual summary.
> Moreover, the usefulness of the presented two-step generation framework lies in the option to control crucial summary properties (discourse structure, style, and content) by providing custom plans. We showcased this situation in Section 6.3, and provided further examples in Appendix F.
>
> Finally, for the sake of completeness, we conducted an ablation study to quantify the impact of each extra signal in the source and the impact of generating a summary conditioned in a content plan.
> Table R.3 presents the performance of $Bart_{plan}$ when the source is ablated of metadata, rhetorical labels, and when no content plan is generated.
> It can be seen that removing each signal has a negative impact in terms of informativeness (ROUGE scores), readability (CLI), and style (Sty).
> This outcome highlights the benefits of using metadata and scientific rhetorical roles in the input, as well as the benefit of our two-step generation framework.
>
>
> | System | R1 | R2 | RL | CLI | Sty |
> | --- | --- | --- | --- | --- | --- |
> | $Bart_{plan}$ | 38.84 | 8.89 | 33.5 | 13.55 | 0.98 |
> | w/o Metadata | 37.16 | 8.07 | 32.48 | 14.37 | 0.96 |
> | w/o Rhet. Labels | 37.06 | 7.96 | 32.25 | 14.18 | 0.94 |
> | w/o Content Plan | 37.77 | 8.52 | 33.06 | 13.84 | 0.95 |
>
> Table R.3. Ablation study of the impact of each extra signal in the input (Metadata, Rhetorical labels) and the impact of generating the plan before the summary (Content Plan).
>
> ## Using only Author and Affiliation Information as Metadata
>
> When enriching the source document with article metadata, we decide to only consider author’s names and affiliations for the following reason.
> Considering the distribution of named entities found in Press Release reference summaries (analyzed in Section 3.2 and depicted in Figure 3), it is worth noting that entities of type Organization and Person are the most frequent entities –after numbers and miscellaneous–.
> Hence, we decided to restrict the usage of metadata in our framework to author’s names and affiliations.
> However, other metadata information was collected, both from the scientific article (e.g. publication venue and year, title) and press release articles (e.g. title, PR publication date, journalistic organization), as detailed in Section 3.1.
> We include the complete metadata in the released dataset so that future investigations can leverage them.

---

### Official Review · Reviewer_qvrA · 2023-08-12

**Soundness:** 5

**Excitement:**

4: Strong: This paper deepens the understanding of some phenomenon or lowers the barriers to an existing research direction.

**Paper Topic And Main Contributions:**

This paper presents a substantial dataset for exploring the process of science journalism, in which scientific findings are presented in accessible terms for a non-specialist audience. The authors carefully extract papers, their abstracts and press releases and associated press release snippets. The dataset is accompanied with analyses that demonstrate the changes in style between these genres. Finally, the authors show that a controlled generation method leveraging Bart is most successful at producing high-quality journalistic summaries using a number of different metrics as well as human evaluation on multiple axes.

**Questions For The Authors:**

The only thing that I find a little alarming is that in Table 6 the proposed system scores lower on factuality than GPT-3.5, which itself is not known for its faithfulness. I know that the Bart system is not the main contribution of the paper, but I think there should be at least a discussion of the apparent tradeoff between quality and factuality, especially since this is arguably a domain in which factuality is very important. Error analysis of the shortcomings in factuality would make a great addition to this paper.

**Reasons To Accept:**

This is an excellent and thorough dataset. The analysis in section 3.2 is meticulous and very informative, and gives a very clear picture of the process which the authors seek to automate. The figures are very illuminating, especially Fig. 4. The set of comparison systems is extensive and includes recently released LLMs including open source options, making the claims in the paper very convincing.

**Reasons To Reject:**

I don't really have any strong reasons to reject. My main concern is about factuality (described in questions section), but I don't think it is a sufficient reason to reject this paper.

**Reproducibility:**

5: Could easily reproduce the results.

**Reviewer Confidence:**

4: Quite sure. I tried to check the important points carefully. It's unlikely, though conceivable, that I missed something that should affect my ratings.

---

> ### Author Rebuttal · Authors · 2023-08-28
>
> Thank you for the comments, we are glad you found our contributions and analysis useful.
> As per your suggestion, we conducted an analysis of factuality errors (see below for details) and will add this analysis to the final version of the paper.
>
> ## Trade-off between factuality and other summary qualities
> There is indeed a clear trade-off between factuality and other summary properties, such as Style or Readability, as can be seen in Table 6.
> We found that this limitation is inherent to our approach, and we leave the improvement in factuality to future work.
>
> ## Analysis of Factuality Errors
> We analyzed the types of factuality errors present in candidate summaries.
> We uniformly sampled 30 instances from the test set and manually annotated their respective reference summary and summaries generated by $Bart_{plan}$ and GPT3.5-Turbo.
>
> We adapt the error taxonomy employed in [0] and consider three categories at the span level: (ii) *Entity-related*, when the span is a named entity (same entity categories considered in Section 3.2.); (ii) *Noun Phrase-related*, when the span is an NP modifier; and (ii) *Other Errors*, such as repetitions and grammatical errors.
> Contrary to [0], the Event-related category is not considered here since the source documents in SciTechNews do not contain events.
> Each error category is further divided into sub-categories: *Intrinsic*, *Extrinsic*, and *World Knowledge*, depending on where the supporting information is found [1].
> Intrinsic errors are caused by phrases or entities found in the source being generated in the wrong place.
> In contrast, extrinsic errors happen when the generated span cannot be supported by the input source or any external source (e.g. Wikipedia).
> Finally, word knowledge errors are caused when the span cannot be verified with the source but it can be verified using external knowledge, e.g. author X being the director of the Computer Science department at university Y.
>
> Table R.2. presents the proportion of error categories found in the inspected summaries, along with the total number of error spans found for each system. It is worth noting that the total number of errors follows the ranking trend in Table 6, with PR summaries having the least number of errors, followed by GPT3.5, and then Bart_{plan}.
> First, we observe that reference summaries exhibit only Entity and NP-related errors of type World Knowledge. The majority of them include completed names of affiliated institutions (e.g. the metadata mentioning only the abbreviation “MIT” but the reference summary showing the complete name), country names where these institutions are located, or the position an author holds within an institution.
> We also found cases where an author held more than one affiliation, with only one of these being listed in the metadata and another mentioned in the reference summary.
> Second, we observe that Bart_{plan} extrinsically hallucinates mostly entities (e.g. country names, 34% of all its errors), followed by extrinsic NPs.
> Among intrinsic errors, entity-related ones included mixed-ups of author first names, last names, and affiliations, whilst NP-related errors included mistaking resources mentioned in the source (e.g. a github repository) for institutions.
> In contrast, GPT3.5-Turbo produced a high proportion of extrinsic hallucinations of noun phrases, most of them stating publication venues (e.g. *In a paper published in …*). Since the metadata added to the source does not include publication venue, the model must have surely been exposed to this kind of information during training. However, somewhat surprisingly, GPT3.5-Turbo did not exhibit world knowledge errors of any kind, potentially highlighting the conservative approach to generation followed during its training.
> Errors of type “Other” consisted mainly of orphaned phrases at the beginning of generation, i.e. the model starts the generation by attempting to continue the last sentence of the input in a `continue the story’ fashion. Consider the following example:
>
> **Input**
> ```
> Write a report of this paper in journalistic style
> Paper:
> [AUTHOR] sachit menon | duke university durham [AUTHOR] … [OBJECTIVE] the primary aim of single - image super - resolution is to construct a … and also downscale correctly . [CONCLUSIONS] the ( weighted ) pixel
> Report:
> ```
> **GPT3.5-Turbo**
> ```
> -wise average distance between the super-resolved and high-resolution images. However, this often results in blurring, especially in detailed regions…
>
> ```
>
> Here, we observe that the input document was truncated in the middle of a sentence, and GPT3.5 attempted to complete it, hallucinating the orphaned span (*-wise average...images.*).
> We hypothesize that the GPT3.5 model employed(model checkpointed at March, 2023) still struggled with the length of the input, reaching a point where the prompt instruction (stated at the beginning of the input) is completely ignored and the model just continues the `story’ given.
>
>
>
> ---
>
> | System | Entity-Int.  | Entity-Ext. | Entity-W.K. | NP-Int. | NP-Ext. | NP-W.K. | Other | Total |
> | ------ | ------ | ---- | ---- | ---- | ---- | ----- | ----- | ----- |
> | PR Summary | 0 | 0 | 0.79 | 0 | 0 | 0.21 | 0 | 43 |
> | $Bart_{plan}$ | 0.1 | 0.34 | 0.20 | 0.07 | 0.16 | 0.02 | 0.11 | 61 |
> | GPT3.5-Turbo | 0 | 0.08 | 0 | 0.02 | 0.18 | 0 | 0.72 | 50 |
>
> Table R.2. Proportion of factuality errors in reference summaries (PR Summary), summaries generated by Bart_{plan}, and summaries generated by GPT3.5-Turbo (bottom). Error categories are organized by Entity-related, Noun Phrase-related (NP), and Other, and further subcategorized into Extrinsic (Ext.), Intrinsic (Int.), and World Knowledge (W.K.). The total is showed by system, with error proportions also normalized by system.
>
>
> ## References
> [0] Goyal, Tanya, and Greg Durrett. "Annotating and Modeling Fine-grained Factuality in Summarization." NAACL-HLT 2021.
>
> [1] Cao, Shuyang, and Lu Wang. "CLIFF: Contrastive Learning for Improving Faithfulness and Factuality in Abstractive Summarization." EMNLP 2021.

---

### Official Review · Reviewer_6Mgm · 2023-08-12

**Soundness:** 4

**Excitement:**

4: Strong: This paper deepens the understanding of some phenomenon or lowers the barriers to an existing research direction.

**Paper Topic And Main Contributions:**

The paper is about automatic science journalism, the task is about generating a news-article-like summary of a scientific paper, the goal is for the summary to look like what a professional journalist would write both in terms of linguistic style and content. The main contributions are a dataset (SciTechNews) with tuples of (scientific paper, corresponding news article, expert-written summary snippet), and a methodology to generate text guided by metadata + discourse structure where the model first generates a plan based on the scientific paper (eg introduction, background, methods) then generates a summary text that follows the plan.

**Reasons To Accept:**

- There is very good motivation for such a research area to exist: given the volume of published scientific work, journalists cannot manually report on all scientific findings, which means an important part of meaningful science is never reported to the general public.
- The framing of the task has some usefulness/originality in that it uses metadata+discourse structure and includes both in the generation process instead of having plain seq2seq models that train only on text content.
- Evaluation against baselines and current LLMs (eg GPT3.5) shows promising results.
- The paper is well written/structured and has good readibility.

**Reasons To Reject:**

- No particular reason to reject the paper.

**Reproducibility:**

3: Could reproduce the results with some difficulty. The settings of parameters are underspecified or subjectively determined; the training/evaluation data are not widely available.

**Reviewer Confidence:**

4: Quite sure. I tried to check the important points carefully. It's unlikely, though conceivable, that I missed something that should affect my ratings.

---

> ### Author Rebuttal · Authors · 2023-08-28
>
> Thanks for the positive review. We really appreciate it.

---

### Meta-Review · Area_Chair_dxV3 · 2023-09-23

**Recommendation:** 4

**Metareview:**

The paper presents a dataset and an automatic summarization method to generate journalistic-style summaries from scientific papers. The method takes advantage of the discursive structure and metadata of the papers to generate the summaries. The method is compared with current LLM-based tools (Alpaca and ChatGPT) showing good results. The paper makes valuable contributions and is well presented.

---

### Decision · Program_Chairs · 2023-10-07

**Decision:**

Accept-Main

**Comment:**

The paper presents a dataset and an automatic summarization method to generate journalistic-style summaries from scientific papers. The method takes advantage of the discursive structure and metadata of the papers to generate the summaries. The method is compared with current LLM-based tools (Alpaca and ChatGPT) showing good results. The paper makes valuable contributions and is well presented.